# Mitochondrial Unfolded Protein Responses in White Adipose Tissue: Lipoatrophy, Whole-Body Metabolism and Lifespan

**DOI:** 10.3390/ijms22062854

**Published:** 2021-03-11

**Authors:** Masaki Kobayashi, Yuichiro Nezu, Ryoma Tagawa, Yoshikazu Higami

**Affiliations:** 1Laboratory of Molecular Pathology and Metabolic Disease, Faculty of Pharmaceutical Sciences, Tokyo University of Science, 2641 Yamazaki, Noda, Chiba 278-8510, Japan; kobayashim@rs.tus.ac.jp (M.K.); 3A15073@ed.tus.ac.jp (Y.N.); tagawar@rs.tus.ac.jp (R.T.); 2Research Institute for Biomedical Sciences, Tokyo University of Science, 2669 Yamazaki, Noda, Chiba 278-8510, Japan

**Keywords:** mitochondrial unfolded protein response (UPR^mt^), mitokine, white adipose tissue (WAT), lipoatrophy, mitohormesis

## Abstract

The mitochondrial unfolded protein response (UPR^mt^) is a stress response mediated by the expression of genes such as chaperones, proteases, and mitokines to maintain mitochondrial proteostasis. Certain genetically modified mice, which defect mitochondrial proteins specifically in adipocytes, developed atrophy of the white adipose tissue, resisted diet-induced obesity, and had altered whole-body metabolism. UPR^mt^, which has beneficial functions for living organisms, is termed “mitohormesis”, but its specific characteristics and detailed regulatory mechanism have not been elucidated to date. In this review, we discuss the function of UPR^mt^ in adipose atrophy (lipoatrophy), whole-body metabolism, and lifespan based on the concept of mitohormesis.

## 1. Introduction

Mitochondria, organelles in eukaryotic cells, originated from specialized aerobic bacteria that were incorporated into the cytoplasm of prokaryotes by endocytosis. The ability of symbiont bacteria to conduct cellular respiration in host cells, which depended on glycolysis and fermentation, would have provided a significant evolutionary advantage. During evolution, DNA of the symbiotic genome was transferred to the nuclear genome of the host cells, but part of it remained as mitochondrial DNA (mtDNA) that replicates autonomously. Mitochondria, which consist of a lipid bilayer membrane containing mtDNA, can replicate and undergo fission and fusion, and unnecessary and/or old damaged mitochondria are removed by autophagy (so-called mitophagy). To maintain mitochondrial quality, host cells have developed various protective systems including the mitochondrial unfolded protein response (UPR^mt^) [1,2,3].

mtDNA encodes 13 proteins that partly make up respiratory chain complexes. Most of the approximately 1200 mitochondrial proteins encoded by nuclear DNA possess an N-terminal mitochondrial targeting sequence, are synthesized in the cytoplasm as precursor forms, and are transferred into mitochondria. After the proteins pass through the mitochondrial double-membranes, the mitochondrial targeting sequences of the precursor forms are cleaved by specific mitochondrial signal peptidases (MtSPases). After the cleavage, the proteins mature and acquire their functions. Mitochondria synthesize ATP by oxidative phosphorylation (OXPHOS) via the respiratory chain complex, citric acid cycle, and β-oxidation. Therefore, mitochondria are called “energy powerplants”. Mitochondria also play important roles in various cellular processes including cholesterol synthesis, amino acid metabolism, heat production, calcium storage, and apoptosis [1,2,3].

White adipose tissue (WAT) is a major tissue for energy storage in the form of triglycerides (TG). Recently, WAT was shown to be an endocrine tissue that secretes adipokines, such as adiponectin, leptin, and various inflammatory cytokines. It is well known that the characteristics of adipocytes and their secretory profiles vary with the size of the adipocyte. In general, large adipocytes, usually observed in obese WAT, accumulate more TG in unilocular lipid droplets and secrete less adiponectin and more leptin and inflammatory cytokines, which leads to inflammation, insulin resistance, and leptin resistance. In contrast, small adipocytes secrete more adiponectin and less inflammatory cytokines, are more sensitive to insulin, and function as a powerful buffer for whole-body lipids by absorbing them after feeding and releasing them before feeding [4].

Currently, it is widely accepted that the growth hormone (GH)/insulin-like growth factor 1 (IGF-1)/insulin signaling pathway is the most important regulator of the lifespan and aging processes [5]. In addition, it was reported that WAT itself plays an important role in the regulation of the lifespan. Systemic *insulin receptor* (*IR*) knockout (KO) mice die shortly after birth due to ketosis [6]. Similarly, pancreatic β-cell-specific *IR* and *IGF-1 receptor* KO mice die early due to severe diabetes [7]. Myocardial and skeletal muscle-specific *IR* and *IGF-1 receptor* KO mice die from heart failure within a few weeks of birth [8]. In addition, liver-specific *IR* KO mice developed severe insulin resistance, leading to progressive liver dysfunction [9]. However, in adipose-tissue-specific *IR* KO mice, the WAT mass was reduced, mitochondrial biogenesis in WAT was enhanced, and their lifespan was extended [10]. Furthermore, transgenic mice, in which the expression of adiponectin was enhanced in the liver, lived longer than controls [11]. CCAAT/enhancer-binding proteins (C/EBPs), sterol regulatory element-binding protein 1 (SREBP1), and peroxisome proliferator-activated receptor γ (PPARγ) are critical transcription factors involved in adipocyte differentiation and maturation [4]. Knock-in mice, in which the *C/EBP*α gene was replaced by the *C/EBPβ* gene (β/β mice), lived longer and showed higher energy expenditure than normal littermates. It was suggested that the beneficial actions of β/β mice might result from increased WAT energy oxidation and the upregulation of mitochondrial proteins [12,13]. In contrast, *Pparγ2* KO mice, which lack PPARγ exclusively in the WAT, developed severe lipodystrophy, remained insulin-resistant throughout life, and died significantly earlier than controls [14]. We previously reported that SREBP1c is a critical transcription factor in caloric restriction (CR)-induced lifespan extension and metabolic remodeling with enhanced mitochondrial biogenesis in the WAT [15]. Thus, the WAT and its mitochondria are involved in the regulation of whole-body metabolism and lifespan.

Here, we discuss the regulation of whole-body metabolism and lifespan based on the characteristics of WAT, its mitochondria, and UPR^mt^.

## 2. Mitochondrial Unfolded Protein Response

The UPR^mt^ is a stress response mediated by the expression of genes that encode chaperones, proteases, and mitokines to maintain mitochondrial proteostasis [1,2]. Mitochondrial chaperones, including mitochondrial heat shock protein 70 (mtHSP70/Mortalin/Grp75), HSP60, and HSP10, are involved in protein folding [2,16,17]. Most nuclear-encoded mitochondrial proteins are imported into mitochondria. During this process, the precursor forms of proteins with a mitochondrial targeting signal enter through the translocase complexes of the outer and inner membranes TOM and TIM, respectively [3]. mtHSP70 is localized in the matrix side and is involved in protein transport into the matrix from the intermembrane space [18]. After the precursor form of matrix proteins enters the mitochondria, the targeting sequence is removed by a mitochondrial processing peptidase, followed by the folding of the imported proteins to their active conformation by the molecular chaperones, HSP60 and HSP10. HSP60 and HSP10 form a symmetrical complex and facilitate mitochondrial protein folding, leading to their stability [17]. mtHSP70 also chaperones the imported proteins to prevent their misfolding, aggregation, and mitochondrial protein degradation [19]. HSP60 also interacts with Survivin. The acute ablation of HSP60 by small interfering RNA destabilized Survivin in mitochondria, induced mitochondrial dysfunction, and activated Bax-dependent apoptosis via p53 stabilization by disruption of the HSP60-p53 complex [20].

The LON protease (LONP1) and CLpXP complex, major proteases in the mitochondrial matrix, are involved in the degradation of unfolded, damaged, and/or toxic proteins. LONP1 plays an important role in the degradation of oxidized proteins in the mitochondrial matrix, particularly after acute stress [2,21]. Tetradecameric CLpPs form the CLpXP complex with hexameric AAA+ chaperone CLpXs. It was reported that ClpX recognizes unstructured peptide sequences in proteins, proceeds to unfold tertiary structures in the proteins, and then translocates the unfolded polypeptide chain into a sequestered proteolytic compartment in ClpP for degradation to small peptide fragments [22].

Growth differentiation factor 15 (GDF15; also known as macrophage inhibitory cytokine-1 (MIC-1), nonsteroidal anti-inflammatory drug-activated gene (NAG1)) and fibroblast growth factor 21 (FGF21) are major UPR^mt^-induced mitokines [1,2]. GDF15 and FGF21 appear to be useful biomarkers in various mitochondrial diseases [23,24,25]. GDF15, a member of the transforming growth factor-beta superfamily, was originally characterized as a macrophage-derived secretory protein [26,27]. It is predominantly expressed in the liver, lung, and kidney in healthy animals [26,28,29], but its expression is also induced in several tissues due to various types of stress [26,30,31,32]. GDF15 binds to a receptor composed of a heterodimer of Ret and a member of the GDNF receptor α family, known as GFRA-like (GFRAL) in some tissues including the hindbrain [26,33]. However, GDF15, its receptor, and downstream signaling are poorly understood. FGF21, a member of the endocrine FGF superfamily, was initially identified as a hepatokine that regulates lipid and glucose metabolism [34,35]. FGF21, which is mostly secreted from the liver into the bloodstream under fasting conditions, binds to the FGF receptor (FGFR) and beta-klotho (KLB) receptor complex in target tissues, such as the WAT and muscle [36,37]. FGF21-FGFR1/KLB signaling is involved in glucose uptake, lipogenesis, and lipolysis [35,36]. Recently, Fgf21 was shown to be expressed in the liver and other tissues, including the WAT, brown adipose tissue, muscle, and pancreas [38]. We recently reported that CR promoted the expression of FGF21 protein expression in the WAT [39].

It is generally accepted that mild-to-moderate UPR^mt^ is beneficial for living organisms (termed “mitohormesis”), whereas severe UPR^mt^ can exacerbate disease. Thus, mitohormesis is defined as “biological responses where the induction of an adequate amount of mitochondrial stress leads to an increment in health and viability within a cell, tissue or organism”[40]. However, its true nature and detailed regulatory mechanism, particularly in mammals, have not been elucidated.

## 3. Genetically Modified Mice Live Longer Due to a Deficiency of Mitochondrial Proteins or Overexpression of Mitokines

It was reported that mice with genetic defects in mitochondrial proteins including CLK-1/MCLK1/COQ7 and SURF1 lived longer than wild mice [41,42]. CLK-1/MCLK1/COQ7 is a mitochondrial hydroxylase involved in ubiquinone (UQ) biosynthesis [43]. Homozygous *Mclk1* KO mice are embryonic lethal, but systemic heterozygous *Mclk1* KO mice are born normally and live longer than controls [41]. In homozygous *Mclk1* KO mice, low ATP levels, high mitochondrial oxidative stress, and low non-mitochondrial oxidative damage were found [43]. Moreover, the mice were associated with the metabolic alteration of mitochondria and were protected against bacterial infection and subsequent tissue damage [44]. *Coq*^Q95X^ mice, which also have defects in UQ biosynthesis, lived longer than controls and had undetectable levels of COQ9 protein with a moderate UQ deficiency in the brain, kidneys, and skeletal muscle [45]. The hepatic UQ levels were not decreased, and mitochondrial dysfunction and increased oxidative stress did not develop in the liver, suggesting tissue-specific differences in UQ biosynthesis. Therefore, the effect of reduced levels of proteins involved in UQ biosynthesis on increased survival of mice may be due to mitochondrial mechanisms in non-liver tissues or other unknown mechanisms [46]. Moreover, another study reported that CLK-1 localized in the nucleus increased the expression of genes involved in mitochondrial reactive oxygen species (ROS) metabolism, leading to a reduction in ROS. In addition, nuclear CLK-1 suppresses UPR^mt^. Reduced ROS levels led to the trafficking of CLK from the nucleus and its predominant localization in mitochondria. This study suggested that nuclear or mitochondrial CLK-1 functions as a rheostat to maintain ROS homeostasis and attenuate UPR^mt^ [47]. Thus, the molecular mechanisms involved in lifespan extension related to defects of CLK-1 require further study (Table 1).

SURF1 is an assembly protein for complex IV (cytochrome c oxidase; COX) in the electron transport chain. Patients with a mutation of the *Surf1* gene suffer defects of COX activity in multiple tissues, resulting in fatal mitochondrial encephalomyelopathy, termed Leigh syndrome. In contrast, mice lacking the SURF1 protein (*Surf1* KO mice) are viable and have a >50% reduction in COX activity, resistance to Ca^2+^-dependent neurodegeneration, and extended longevity [42]. *Surf1* KO mice also had a lower body and fat mass, in association with reduced lipid storage, smaller adipocytes, and elevated fatty acid oxidation in the WAT compared with control mice. The respiratory quotient in *Surf1* KO mice was significantly lower than that in control animals, and elevated fat utilization was associated with enhanced glucose metabolism. The expression of peroxisome proliferator-activated receptor γ-coactivator 1-α (PGC-1α) and its target genes was upregulated in the WAT, heart, and skeletal muscle of *Surf1* KO mice [48]. Despite the significant reduction in COX activity, there was little or no difference in ROS generation, membrane potential, ATP production, or respiration in isolated mitochondria from the heart and skeletal muscle of *Surf1* KO mice compared with control mice. In addition, the UPR^mt^-associated proteins, HSP60, ClpP, and LONP1 were elevated in skeletal muscle but not in the heart of *Surf1* KO mice compared with control mice, suggesting a tissue-specific effect of UPR^mt^ in response to SURF1 deficiency [49,50]. Taken together, these findings suggest that SURF1 deficiency promotes fat utilization, enhances insulin sensitivity, activates mitochondrial biogenesis in various tissues, induces UPR^mt^ in the heart and skeletal muscle, and extends longevity (Table 1).

Mitokines, including GDF15 and FGF21, are also involved in whole-body metabolism and/or lifespan [2]. *Fgf21* transgenic (Tg) mice, in which the Fgf21 transgene is selectively expressed in hepatocytes under the control of the ApoE promoter, live markedly longer than controls without reducing food intake. *Fgf21* Tg mice had 5–10 times higher circulating concentrations of FGF21 than controls. Younger *Fgf21* Tg mice had significantly decreased insulin, IGF-1, glucose, TG, and cholesterol levels in the serum and TG levels in the liver. These findings based on hepatic transcriptomic analysis suggest that the significant extension of lifespan in *Fgf21* Tg mice might be a result of blunting the GH/IGF-1 signaling pathway in the liver rather than affecting NAD^+^ metabolism, AMP kinase, or mTOR signaling [51].

Female *Gdf15* Tg mice had a smaller body size and lived markedly longer than controls. Furthermore, *Gdf15* Tg mice had increased serum GH levels but decreased levels of serum IGF-1, insulin, and leptin. Insulin sensitivity and energy expenditure were increased with higher lipid mobilization [52]. In male *Gdf15* Tg mice, thermogenesis and oxidative metabolism were increased, glucose tolerance was improved [53], and inflammation was inhibited [54]. Macrophage-specific *Crif1* KO mice developed insulin resistance with M1 macrophage-like polarization, but GDF15 treatment upregulated the oxidative function of macrophages, leading to their polarization to an M2-like phenotype, which reversed insulin resistance in *Crif1* KO mice fed an HFD, suggesting GDF15 improved glucose metabolism with altered macrophage polarization in the WAT [55]. Taken together, GDF15 suppresses insulin/IGF-1 signaling and inflammation with polarization to an M2 macrophage-like phenotype and enhances energy expenditure with the activation of lipid mobilization and thermogenesis, leading to lifespan extension and suppression of age-related pathologies.

## 4. Genetically Modified Mice Have Defective Mitochondrial Protein, Leading to Lipoatrophy, Which Regulates Whole-Body Metabolism

Mitochondria transcription factor A (TFAM) is important for the stability of mtDNA and initiation of the transcription of genes encoding mtDNA [56]. Adipoq-*Tfam* KO mice, generated by mating with Adiponectin-Cre mice, developed lipoatrophy and had reduced mitochondrial biogenesis and deterioration of glucose metabolism and heart function and increased inflammation in the WAT and fatty liver [57]. However, adipose-specific *Tfam*-deficient (aP2-*Tfam* KO) mice, generated by mating with aP2-Cre mice, developed lipoatrophy and reduced mitochondrial biogenesis but had resistance to diet-induced obesity (DIO), as well as improved glucose metabolism and fatty liver and increased energy expenditure [58]. Unfortunately, markers involved in UPR^mt^ have not been investigated in Adipoq-*Tfam* KO mice or aP2-*Tfam* KO (Table 1).

CR6-interacting factor 1 (CRIF1), also known as CR6/GADD45-interacting protein, is an essential protein for the intramitochondrial translation of mtDNA-encoded oxidative phosphorylation (OXPHOS) subunits [59]. Therefore, CRIF1 deficiency reduced OXPHOS activity. Fat-specific Crif1 deficient (Adipoq-*Crif1* KO) mice, generated by mating with Adiponectin-Cre mice, developed lipoatrophy, but whole-body metabolism was not altered when fed a normal diet (ND). In Adipoq-*Crif1* KO mice, glucose metabolism was improved, and energy expenditure was increased under the conditions of a high-fat diet (HFD). Moreover, the expression levels of proteins involved in mitochondrial chaperones and proteases, and mRNAs of mitokines, Gdf15 and Fgf21, were upregulated in the WAT of Adipoq-*Crif1* KO mice fed ND and HFD. The deletion of either Gdf15 or Fgf21 partly canceled the beneficial effects observed in Adipoq-*Crif1* KO mice. Thus, the characteristics seen in Adipoq-*Crif1* KO mice might be related to UPR^mt^-induced GDF15 and/or FGF21 expressions [60] (Table 1).

As described above, ClpP, a component of caseinolytic peptidase (ClpXP), is a proteolytic compartment for the degradation of proteins into small peptide fragments as mentioned above [2,22]. *ClpP* KO mice showed reduced adiposity, improved insulin sensitivity, and resistance to DIO. Mitochondrial biogenesis was enhanced selectively in the WAT but not brown adipose tissue (BAT), heart, or skeletal muscle of mice fed ND and HFD. UPR^mt^ markers, including LONP1 and mitochondrial chaperones including HSP60, HSP40, HSP10, and ClpX, were upregulated in the WAT of KO mice fed ND. Moreover, KO mice were protected from glucose intolerance, insulin resistance, hepatic steatosis, and increased energy expenditure. Thus, the beneficial effects appeared to be increased for mice fed an HFD [61] (Table 1).

Mitochondrial intermediate peptidase (MIPEP), an MtSPase in the mitochondria matrix, cleaves eight amino acids from the N-terminal via mitochondrial processing peptidase (MPP). Thus, Mipep contributes to the second of two successive cleavages, resulting in the maturation of substrate proteins [3]. We previously reported that Sirtuin 3 (SIRT3), cytochrome c oxidase subunit 4 (COX4), and malate dehydrogenase 2 (MDH2) are substrates for MIPEP in the WAT of mice [62]. However, little is known about MIPEP functions, including its substrate in WAT. To investigate the functions of MIPEP, we generated fat-specific *Mipep*-deficient (Adipoq-*Mipep* KO) mice, generated by mating with Adiponectin-Cre mice. Adipoq-*Mipep* KO developed severe lipoatrophy and were resistant to DIO, but whole-body metabolism was not significantly exacerbated under the conditions of feeding with ND and HFD. In the WAT of Adipoq-*Mipep* KO mice, GDF15 and FGF21 were upregulated, but factors involved in mitochondrial chaperones and proteases were not, suggesting the characteristics seen in Adipoq-*Mipep* KO mice are not typical of UPR^mt^ but are rather an inducible mitokine response (unpublished data, Table 1).

In the BAT of all five genetically modified lipoatrophic mice, which have defective mitochondrial proteins as described in Table 1, reduced size and/or WAT-like phenotypes, including the decreased expression of brown adipocyte markers such as uncoupling protein 1, were found. These findings suggest that enhanced energy expenditure does not occur in the BAT of these lipoatrophic mice [57,58,59,60,61].

## 5. Discussion: Lessons Learned from Genetically Modified Mice that Develop Lipoatrophy Due to Defective Proteins Involved in Mitochondrial Function

Genetic (inherited, congenital) and acquired lipoatrophy in humans, which can be localized or generalized in distribution, is a heterogenous syndrome. Despite its pathogenesis and different distributions, it is often associated with various metabolic disorders and tissue damage, including insulin resistance, dyslipidemia, and ectopic lipid accumulation such as fatty liver. In these cases, it is termed lipodystrophy. The pathogenesis of human lipodystrophy can be roughly characterized as defective adipocyte differentiation, mitochondrial dysfunction including antiretroviral treatment, dysregulation of lipid metabolism, defective DNA damage repair, or abnormal LMNA genes, including the model for Hutchinson-Gilford Progeria Syndrome, termed “laminopathy” [63]. Thus, mitochondrial dysfunction is one pathogenic form of lipodystrophy.

PPARγ is a master regulator of adipocyte differentiation [4]. Adipose-specific *Pparγ*-deficient (aP2-*Pparγ* KO) mice, generated by mating with aP2-Cre mice, developed lipoatrophy with suppressed adipocyte differentiation and fatty liver. Glucose metabolism was not changed under the conditions of ND feeding. Even in aP2-*Pparγ* KO fed an HFD, glucose metabolism was slightly exacerbated with an approximately two-fold increase in plasma insulin levels [64]. A different adipose-specific *Pparγ*-deficient (Adipoq-*Pparγ* KO) mouse, generated by mating with Adiponectin-Cre mice, developed lipoatrophy with suppressed adipocyte differentiation. Glucose metabolism was markedly exacerbated with a >100-fold increase in serum insulin levels. In addition, significant inflammatory cell infiltration and fibrosis were observed in the WAT [65]. In general, serum or plasma insulin levels are elevated relative to the severity of glucose intolerance. Glucose metabolism in Adipoq-*Pparγ* KO mice was markedly exacerbated compared with that in aP2-*Pparγ* KO mice. Taken together, it is likely that glucose metabolism exacerbated in genetically modified lipoatrophic mice resulting from a deficiency of TFAM or PPARγ is more likely to occur on a background of Adiponectin-Cre than aP2-Cre. Moreover, *aP2* is a highly expressed gene in adipocytes but is also expressed in other tissues and cells including macrophages [66]. Indeed, Cre-recombinase activity was present in various tissues in aP2-Cre mice [67]. According to these findings, it is widely accepted that Cre-recombinase activity is more significant and specific in adipocytes from Adiponectin-Cre mice than from aP2-Cre mice. Therefore, mitochondrial stress in WAT is more severe in KO mice in an Adiponectin-Cre background than in aP2-Cre background.

Recently, it was reported that the inhibition of mitochondrial translation led to the coordinated suppression of cytosolic translation, which might be targeted to promote lifespan [68]. Therefore, we should examine the mito-cytosolic translational balance to understand the pathologies involved in UPR^mt^. From the viewpoint of glucose tolerance and UPR^mt^ based on the findings described in this review, we would like to propose “mitohormesis” in lipoatrophic models (Figure 1). The various severities of mitochondrial stress resulting from defective proteins involved in mitochondrial function induce varying degrees of UPR^mt^. In cases of mild mitochondrial stress such as that in Adipoq-*Crif1* KO mice [60], in Cag-*Clpp* KO mice [61], and in aP2-*Tfam* KO mice [58], UPR^mt^, including chaperones, proteases, and mitokines, might be fully induced, probably improving whole-body metabolism including glucose tolerance. Such mild UPR^mt^ might extend the lifespan. To the best of our knowledge, UPR^mt^ has not been evaluated in *Clk1* KO mice, which have high mitochondrial oxidative stress, low non-mitochondrial oxidative damage, and a long lifespan; however, it was reported that mild UPR^mt^ was induced in *Surf1* KO mice, which also have a long lifespan [41,42]. Therefore, the mitochondrial oxidative stress might also induce UPR^mt^. GDF15 and FGF21 are representative mitokines induced by UPR^mt^. As mentioned above, *Fgf21* Tg and *Gdf15* Tg mice have extended lifespans [51,52]. In the case of our Adipoq-*Mipep* KO mice, the expressions of mitokines were induced, but those of mitochondrial chaperones and proteases were not. Therefore, it is likely that UPR^mt^ is not fully induced in these mice or that UPR^mt^-induced beneficial actions are canceled by deteriorated mitochondrial functions. Unfortunately, UPR^mt^ has not been investigated in cases of severe mitochondrial stress such as that in Adipoq-*Tfam* KO or Adipoq-*Pparγ* KO mice. However, UPR^mt^ might be not induced, or severe mitochondrial stress might surpass the beneficial actions of UPR^mt^, leading to accelerated glucose intolerance, ectopic lipid accumulation, and inflammation (Figure 1).

Finally, to further understand the pathology and/or beneficial actions of UPR^mt^ in lipoatrophic animal models, it is important to evaluate the severity of glucose tolerance and the degree of adipocyte differentiation by measuring plasma insulin, plasma adiponectin, and leptin levels. In addition, the importance of measuring plasma GDF15 and FGF21 levels as biomarkers of UPR^mt^ should be highlighted.

## Figures and Tables

**Figure 1 ijms-22-02854-f001:**
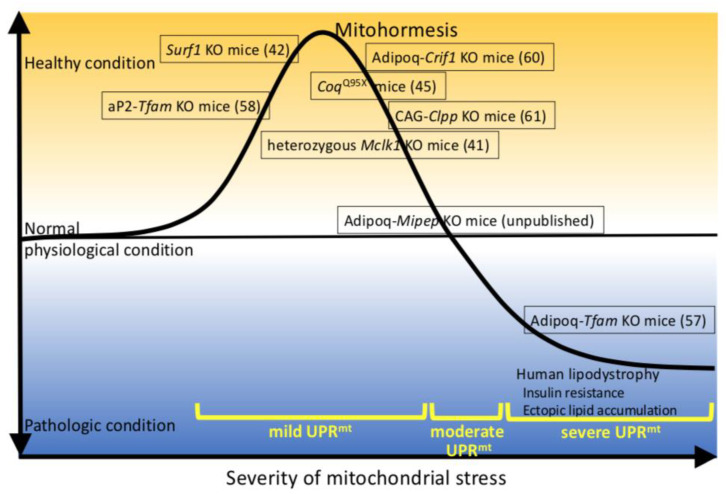
Proposed mechanism of interaction between the severity of mitochondrial stress and healthy conditions in mice based on findings derived from genetically modified mice, which develop lipoatrophy due to defective proteins involved in mitochondrial function. Numbers in parentheses indicate references.

**Table 1 ijms-22-02854-t001:** Genetically modified mice, which show lipoatrohy due to the defect of proteins involved in mitochondria function.

Mice Genotype	Function of Target Protein	Body Weigh	Alteration of WAT and Adipocytes	Alteration Of Mitochondria Function	Glucose Metabolism	Upr^mt^	Adipocyte Differ.	Others	Reference
Genetically modified mice, which live longer due to deficiency of mitochondrial proteins									
Heterozygous *Mclk1* KO	Ubiquinone biosynthesis	→	n.r.	Complex II activity in ES cells ↓ Complex II, Complex I-III, Complex II-III activity in liver →	n.r.	n.r.	n.r.	・Resistant to oxidative stress and infection ・Enhanced immune response	[41,43,44]
*Coq* ^Q95X^	Ubiquinone biosynthesis	↓	n.r.	Proteins involved in ubiquinone biosynthesis in brain, heart, kidney and SKM ↓, in liver → Complex I-III, Complex II-III activity in female kidney, SKM ↓, in brain →	n.r.	Factors involved in chaperone and protease in liver →	n.r.	・oxidative stress → ・Voluntary running distance ↓	[45,46]
*Surf1* KO	Complex IV assembly	↓	WAT weight: ↓ Adipocyte size: ↓	Complex IV activity in liver, heart, SKM and WAT ↓ Complex I, II, III activity in heart, SKM → Mitochondria biogenesis in WAT ↑ Amount of Complex II, V in WAT ↑	Improved	Factors involved in chaperone and protease in liver, heart and/or SKM↑	↑/→	・Plasma insulin ↓ ・Resistant to Ca^2+^- dependent neurodegeneration	[42,48,49,50]
Genetically modified mice, which defect mitochondrial protein, induce lipoatrophy									
*Tfam* KO (aP2-Cre)	Stabilization and transcription of mtDNA	↓	WAT weight: ↓ Adipocyte size: ↓	mtDNA derived factors in WAT ↓ CS in WAT ↑ Complex I, IV activity in WAT ↓	ND: improved HFD: improved	n.r.	n.r.	・Plasma insulin ↓	[58]
*Tfam* KO (Adipoq-Cre)	Stabilization and transcription of mtDNA	↓	WAT weight: ↓ Adipocyte size: →	Factors encoded mtDNA in WAT ↓ CS activity in WAT ↑ OXPHOS subunits encoded nuclear DNA in WAT ↓ Complex I, II-III, IV activity in WAT ↓	ND: worsen HFD: worsen	n.r.	n.r.	・Fatty liver ・Inflammation ・Hypertension ・Cardiac dysfunction ・Plasma insulin ↑	[57]
*Crif1* KO (Adipoq-Cre)	Translation in mitochondria	↓	WAT weight: ↓ Adipocyte size: ↓	OXPHOS formation in WAT ↓	ND: → HFD: improved	GDF15, FGF21 ↑ Factors involved in chaperone and protease ↑	→	・Macrophagein WAT (predominantly M2) ↑ ・Fatty liver ↓	[60]
*Clpp* KO (Cag-Cre)	Mitochondrial protease	↓	WAT weight: ↓ Adipocyte size: ↓	Mitochondria biogenesis in WAT ↑ βoxidation in WAT ↓ OXPHOS Complex Ⅱ in WAT ↑	ND: improved HFD: markedly improved	Factors involved in chaperone and protease ↑	→	・Plasma insulin ↓ ・Thermogenesis ↑	[61]
*Mipep* KO (Adipoq-Cre)	Mitochondrial signal peptidase	↓	WAT weight: ↓ Adipocyte size: →	Proteins encoded nuclear DNA (COX4, SIRT3) in WAT ↓ Proteins encoded nuclear DNA (MDH2) in WAT ↑ CS activity in WAT →	ND: worsen HFD: slightly worsen	GDF15, FGF21 ↑ Factors involved in chaperone and protease ↓	↓	・Plasma insulin ↑	unpublished

ES: embryonic stem cell; SKM: skeletal muscle, WAT: white adipose tissue; CS: citrate synthase; OXPHOS: oxidative phosphorylation; n.r.: not reported. COX4: cytochrome c oxidase subunit 4; SIRT3, MDH2: malate dehydrogenase 2; differ: differentiation; →: no change; ↓: decrease; ↑: increase.

## Data Availability

Not applicable.

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
