# Peer review of "Mitochondrial Unfolded Protein Responses in White Adipose Tissue: Lipoatrophy, Whole-Body Metabolism and Lifespan"

_ijms, 2021, doi:10.3390/ijms22062854_

Round 1

Reviewer 1 Report

In the review entitled "Mitochondrial unfolded protein responses in white adipose tissue: lipoatrophy, whole-body metabolism and lifespan" they address a complex topic, in which many metabolic pathways and factors are involved, however the authors make a review in which they clearly address the most significant results of the subject and describe the controversy that appears in the different studies.

I think the review is well planned, the approach is clarifying and in a simple way they describe how alterations in WAT are involved in metabolic changes that ultimately lead to lipoatrophy and changes in lifespan.

Some comments for authors are:

-Line 54-55: the authors use an outdated reference (5) to make a current claim. I suggest an update to the reference.

-Line 89: After “HSP60 and” it is necessary to put HSP10, the name of the other chaperone.

-Line 141: it would be appropriate to indicate which are the defects in UQ biosynthesis of the model to which they refer, to be able to compare the results that they stand out.

-Line 145: in the reference 45 did not report a decrease in UQ levels, it would be convenient to change the statement to “reduced levels of proteins involved in UQ biosynthesis”, or similar.

-Line 173: Reference 49 does not appear in table 1, to correct in the table or in the text.

-Line 181: references 50, 51 and 52 do not appear in table 1, to correct in the table or in the text.

-Lines 210 and 214: indicate changes in glucose metabolism such as improved insulin sensitivity, glucose intolerance and insulin resistance, but in the table 1 it is indicated that there are no changes with ND. It is necessary to correct this discordance.

-Table 1: it would be convenient to include in the table 1 the third experimental model that is commented on in the section "genetically modified mice, which live longer due to deficiency of mitochondrial proteins". In the text this model is continuously compared with the Mclk1 KO model and it would be appropriate to see in the table the differences between both.

-Table 1: correct in the Crif1 kO model row the cell corresponding to “others” since the first and last rows are not seen properly.

Author Response

Reviewer 1

Thank you for your positive evaluation.

Please find our point by point response to the comments raised by reviewer #1 as the followings.

1) Line 54-55: the authors use an outdated reference (5) to make a current claim. I suggest an update to the reference.

1) Thank you for your pointing out. We mistook reference number 5 to 6. We corrected reference number and selected recent publication for reference number 5.

2) Line 89: After “HSP60 and” it is necessary to put HSP10, the name of the other chaperone.

2) Thank you for your pointing out our mistake. We corrected the sentence.

3) Line 141: it would be appropriate to indicate which are the defects in UQ biosynthesis of the model to which they refer, to be able to compare the results that they stand out.

8) Table 1: it would be convenient to include in the table 1 the third experimental model that is commented on in the section "genetically modified mice, which live longer due to deficiency of mitochondrial proteins". In the text this model is continuously compared with the Mclk1 KO model and it would be appropriate to see in the table the differences between both.

3 & 8) Thank you for your great suggestion. We distinguished and explained separately the two mice strains, heterozygous Mclk KO mice and CoqQ95X mice in line130 – 141 in a new version of our manuscript. We also added the new column for CoqQ95X mice in Table 1.

4) Line 145: in the reference 45 did not report a decrease in UQ levels, it would be convenient to change the statement to “reduced levels of proteins involved in UQ biosynthesis”, or similar.

4) Thank you for your pointing out. We corrected the sentence in line 140 in a new version of our manuscript as reviewer #1 recommended.

5) Line 173: Reference 49 does not appear in table 1, to correct in the table or in the text.

6) Line 181: references 50, 51 and 52 do not appear in table 1, to correct in the table or in the text.

5 & 6) Thank you for your pointing out the mistake. We deleted the word “Table 1” in line 175and line 189 in a new version of our manuscript.

7) Lines 210 and 214: indicate changes in glucose metabolism such as improved insulin sensitivity, glucose intolerance and insulin resistance, but in the table 1 it is indicated that there are no changes with ND. It is necessary to correct this discordance.

7) Thank you for your pointing out our mistake. We corrected in Table 1.

9) Table 1: correct in the Crif1 kO model row the cell corresponding to “others” since the first and last rows are not seen properly.

9) Thank you for your pointing out. We corrected the column height in the cell in Table 1.

Reviewer 2 Report

This review mentions a concept of current and past studies into the cell-autonomous and cell-nonautonomous endocrine effects of mitochondrial unfolded protein response originated from adipose tissue.  This is a very informative and timely review. The review is well- conceived, clearly written, and coherent. The narrative sometimes reads like a list of the conclusions of published papers without much in the way of their critical evaluation. It would therefore be nice to understand the authors’ views on whether this knowledge casts significant doubts or raises questions about particular published findings and how any puzzling discrepancies might be best clarified.

Minor comments

  1. Lipoatrophy used for the clinical description on the localized loss of fat tissue esp. subcutaneous tissue. I understand the mice model of UPRmt described in this manuscript showed reduced fat mass on normal chow or high fat diet, but the adipose phenotype of the most of adipose UPRmt mice were not same as the human lipoatrophy. I would like to recommend clarify the role of lipoatrophy from the role of UPRmt in the different mice model (Fig.1).
  2. The most of mice described in this manuscript has the phenotypes in brown adipose tissue. Please discuss the possible role of brown adipose tissue in discussion section.  

Line 256-257, The previous studies (PMID: 23516375) showed that Crif1 haploinsufficient deficiency in mice under the control of aP2-cre driver results normal growth and development, they became insulin-resistant. aP2-cre activity was observe in macrophages. It is necessary to discuss the different effect of aP2-cre and adiponectin-cre was not only depend on the recombinase activity. In addition, recent observations (PMID: 29674655) show that GDF15-stimulated macrophages regulate adipose inflammation and improve systemic insulin resistance. The inclusion of said article would strengthen the relationship between GDF15 and regulation of adipose glucose metabolism.

Author Response

Reviewer 2

Thank you for your positive evaluation.

Please find our point by point response to the comments raised by reviewer #2 as the followings.

1) Lipoatrophy used for the clinical description on the localized loss of fat tissue esp. subcutaneous tissue. I understand the mice model of UPRmt described in this manuscript showed reduced fat mass on normal chow or high fat diet, but the adipose phenotype of the most of adipose UPRmt mice were not same as the human lipoatrophy. I would like to recommend clarify the role of lipoatrophy from the role of UPRmt in the different mice model (Fig.1).

1) We completely agree the comment. We added the word “human lipodystrophy” in Figure 1.

2) The most of mice described in this manuscript has the phenotypes in brown adipose tissue. Please discuss the possible role of brown adipose tissue in discussion section.

2) We added the comment of BAT about lipoatrophic models from line 237 to 241 in a new version of our manuscript as reviewer #2 recommended.

3) Line 256-257, The previous studies (PMID: 23516375) showed that Crif1 haploinsufficient deficiency in mice under the control of aP2-cre driver results normal growth and development, they became insulin-resistant. aP2-cre activity was observe in macrophages. It is necessary to discuss the different effect of aP2-cre and adiponectin-cre was not only depend on the recombinase activity. In addition, recent observations (PMID: 29674655) show that GDF15-stimulated macrophages regulate adipose inflammation and improve systemic insulin resistance. The inclusion of said article would strengthen the relationship between GDF15 and regulation of adipose glucose metabolism.

3) Thank you for your great suggestion. We added the comment from line 181 to 185 and line 266 to 268 in a new version of our manuscript as reviewer #2 recommended.